# Citation Network Analysis of the Novel Coronavirus Disease 2019 (COVID-19)

**DOI:** 10.3390/ijerph17207690

**Published:** 2020-10-21

**Authors:** Clara Martinez-Perez, Cristina Alvarez-Peregrina, Cesar Villa-Collar, Miguel Ángel Sánchez-Tena

**Affiliations:** School of Biomedical and Health Science, Universidad Europea de Madrid, 28670 Madrid, Spain; cristina.alvarez@universidadeuropea.es (C.A.-P.); villacollarc@gmail.com (C.V.-C.); miguelangel.sanchez@universidadeuropea.es (M.Á.S.-T.)

**Keywords:** pandemic, COVID-19, public health, SARS-CoV-2, pneumonia

## Abstract

Background: The first outbreaks of the new coronavirus disease, named COVID-19, occurred at the end of December 2019. This disease spread quickly around the world, with the United States, Brazil and Mexico being the countries the most severely affected. This study aims to analyze the relationship between different publications and their authors through citation networks, as well as to identify the research areas and determine which publication has been the most cited. Methods: The search for publications was carried out through the Web of Science database using terms such as “COVID-19” and “SARS-CoV-2” for the period between January and July 2020. The Citation Network Explorer software was used for publication analysis. Results: A total of 14,335 publications were found with 42,374 citations generated in the network, with June being the month with the largest number of publications. The most cited publication was “Clinical Characteristics of Coronavirus Disease 2019 in China” by Guan et al., published in April 2020. Nine groups comprising different research areas in this field, including clinical course, psychology, treatment and epidemiology, were found using the clustering functionality. Conclusions: The citation network offers an objective and comprehensive analysis of the main papers on COVID-19 and SARS-CoV-2.

## 1. Introduction

Coronaviruses (Coronaviridae) are a family of viruses that cause infections in humans and animals. It is a zoonotic disease, i.e., it can be transmitted from animals to humans through direct contact with infected animals or their secretions [1,2]. In 2003, SARS-CoV-1 caused almost 8000 cases in 27 countries and had a 10% mortality rate. Between that year and December 2019, no further cases were detected in humans [3]. Then, a group of pneumonia cases with an unknown origin was reported in Wuhan, in the Chinese province of Hubei, on 8 December 2019 [4]. On 7 January 2020, the Chinese authorities identified a new outbreak of the Coronaviridae family after analyzing several samples from the respiratory tract, which was subsequently named SARS-CoV-2 [4,5]. On 11 February, the General Director of the WHO (World Health Organization) announced “COVID-19” as the name of the disease caused by SARS-CoV-2. A month later, the WHO declared the pandemic status following an increase in the number of cases, which had reached 118,000 infections and more than 4000 deaths in 114 countries [6].

In Europe, the first case was detected in France on 24 January, and the virus then spread to nine other countries in just one month (Belgium, Finland, France, Germany, Italy, Russia, Spain, Sweden and the United Kingdom). By March, the number of positive cases in Europe had increased to 195,511 [7]. Meanwhile, between 21 January and 28 February, 33 positive cases were reported in the Americas in four different countries: the United States (15 cases), Canada (15 cases), Brazil (1 case) and Mexico (2 cases) [8].

Currently, according to WHO data, from the outbreak of the disease to 28 July 2020, the number of positive cases detected globally is 16,341,920 (Africa: 726,105; Americas: 8,728,962; Eastern Mediterranean: 1,494,697; Europe: 3,261,042; South-East Asia: 1,838,380; and Western Pacific: 291,993) [9].

Regarding the symptomatology, the most frequent symptoms are fever (87.9%), dry cough (67.7%), asthenia (38.1%), expectoration (33.4%), dyspnea (18.6%), a sore throat (13.9%), headache (13.6%), myalgia or arthralgia (14.8%), chills (11.4%), nausea or vomiting (5%), nasal congestion (4.8%), diarrhea (3.7%), hemoptysis (0.9%) and conjunctival congestion (0.8%). Other less frequent symptoms that have been identified are neurological, cardiac, ophthalmological, dermatological, hematological and ENT (ear, nose and throat)-related alterations [10].

The human-to-human transmission mechanism is based on direct contact with respiratory droplets of more than 5 microns, as well as via the hands and contact with the mucosa of the mouth, nose or eyes, as SARS-CoV-2 has been detected in nasopharyngeal secretions [11,12]. Additionally, it has been reported that SARS-CoV-2 remains on copper, cardboard, stainless steel and plastic surfaces for 4, 24, 48 and 72 h, respectively, at 21–23 degrees (Celsius) and 40% humidity [13]. On wood, clothes or glass, it can remain for 1–2 days, and it can persist for more than 4 days on stainless steel, plastic or banknotes [14].

Given the absence of a successful treatment, the antimalarial chloroquine and the analogous compound hydroxychloroquine, which have also been used in cases of autoimmune diseases, have been choices for treatment. It has been demonstrated that these medicines have antiviral properties and also immunomodulatory effects [15,16]. Furthermore, there have been attempts to create an S-protein-based vaccine that would induce antibodies that would prevent subsequent infections caused by the wild-type virus [17,18].

Analysis through citation networks is used to search scientific literature on a specific subject. In other words, a publication can be used to find other relevant publications to demonstrate, qualitatively and quantitatively, the relationships between articles and authors through the creation of groups [19]. Furthermore, it is possible to quantify the most cited publications in each group, as well as study the development of a research area or focus the literature search on a specific subject [20,21].

Consequently, considering the increasing number of publications on COVID-19, this study aims to identify the different research areas and to determine the most frequently cited publication. Furthermore, it aims to analyze the relationships between publications and different research groups using the CitNetExplorer software (Centre for Science and Technology Studies (CWTS), Leiden, The Netherlands), which examines the development of the scientific literature in a given research field.

## 2. Materials and Methods

### 2.1. Data Source

The search of publications was carried out using the Web of Science (WOS) database with the following search terms: “COVID-19”, “SARS-CoV-2”, “The Coronavirus Disease 2019” and “Corona Virus Disease 2019”. These terms were selected bearing in mind the study objective; additionally, these are the most common terms in all research fields. Since the results of the search comprised some articles shared by several fields, the Boolean operator NOT was used in the second search (“SARS-CoV-2” NOT “COVID-19”), the third search (“The Coronavirus Disease 2019” NOT “SARS-CoV-2” NOT “COVID-19”) and the fourth search (“Corona Virus Disease 2019” NOT “The Coronavirus Disease 2019” NOT “SARS-CoV-2” NOT “COVID-19”). Additionally, the search was carried out by selecting “Subject” as the search field, and it was limited by abstract, title and keywords. In this way, we covered all articles that included other terms, such as “2019 novel coronavirus”, “2019 Novel Coronavirus Diseases” or “2019-nCoV”. The period selected was from January 2020 to July 2020.

Furthermore, Web of Science makes it possible for users to add references to their library when conducting bibliographic searches directly in external databases or library’ catalogs. With regards to the citation indexes, Social Sciences Citation Index, Science Citation Index Expanded and Emerging Sources Citation Index were used. On the other hand, given that the way in which authors and institutions cite works may vary, the CiteSpace software was used to standardize the data. The date on which the publications were searched and downloaded was 28 July 2020.

### 2.2. Data Analysis

The publications were analyzed using the Citation Network Explorer software, which allows the researcher to analyze and visualize citation networks of scientific publications. Furthermore, it is possible to download citation networks directly from Web of Science and manage citation networks that include millions of publications and related citations. Therefore, a citation network composed of several millions of publications can be the starting point from where a deeper analysis can be conducted to obtain a small subnetwork with 100 publications on the same subject. A quantitative analysis of the most mentioned publications in a period was carried out using the attribute citation score. Therefore, not only the internal connections within the Web of Science database but also any external connections were quantified, meaning that other databases were considered [21]. CitNetExplorer provides several techniques for the analysis of publication citation networks. The clustering functionality is achieved using the formula developed by Van Eck in 2012 (Equation (1)) [21].
(1)$$V\left(c_{1},\ldots ,c_{n}\right)=\underset{i<j}{\sum }\delta \;\left(c_{i},c_{j}\right)\left(s_{ij}-\gamma \right)$$

Then, to assign a group to each publication, the clustering functionality was applied. As a result, the most related publications are usually found in the same group based on citation networks [21].

Finally, the core publications were analyzed using the identifying core publications functionality, which consists of identifying the publications that are considered the core of a citation network; that is, they have a minimum number of connections with other core publications, so those that are irrelevant may be eliminated. The number of connections is established by the researchers, so the higher the value of this parameter, the lower the number of core publications [21]. In this study, the publications that have four or more citations in the citation network were considered. Additionally, the drilling down functionality, which allows for a deeper analysis of each of the groups at different levels, was also used. The VOSviewer software (Centre for Science and Technology Studies (CWTS), Leiden, The Netherlands) was used to create and visualize bibliometric networks (journals, researchers, individual publications, bibliographic relationships, etc.) through graphs.

## 3. Results

The first articles on COVID-19 were published at the beginning of 2020, so the period of time selected was from January 2020 to July 2020. A total of 14,409 publications and 42,377 citation networks were found in the search in WOS. Among all the publications, 39.90% were articles, 26.42% were editorial material, 18.73% were letters, 9.93% were reviews, 4.3% were news items and 0.8% were edits, book reviews, retractions or meeting abstracts. Figure 1 shows the publications with greater weights and the group to which they belong. The color of an article represents its group, and the lines that connect the elements represent links.

The number of publications on COVID-19 has increased since April 2020 (January–March 2020: 2.21% of the publications; April–December 2020: 97.79% of the publications). June was the month with the largest number of publications, with 4212 publications and 1990 citation networks (Figure 2).

Table 1 shows the 20 most cited publications in this citation network. The most cited article is that of Guan et al. [22], published in April 2020, with a citation index of 1150. In this publication, 1099 patients were included from 19 December 2019 (when the disease first broke out) to 29 January 2020. The patients were from 552 hospitals in 30 different regions of Mainland China, out of which 67 were excluded because they had been admitted to the ICU (intensive care unit), were on invasive mechanical ventilation or had died.

The most common symptoms were fever (43.8% when admitted to hospital and 88.7% during hospital stay), cough (67.8%) and diarrhea (3.8%). In addition, ground-glass opacity of the lung was the most common radiological finding in thorax computed tomography (CT) scans (56.4%), and 83.2% of patients presented lymphocytopenia upon admission. The average hospitalization time was 4 days.

During the first two months of the outbreak, COVID-19 spread quickly throughout China. The disease often presented with different degrees of severity, as patients frequently showed no signs of fever or unusual radiological findings.

When analyzing the 20 most cited articles, 17 can be found on the symptoms, viral transmission and experimental treatment methods [22,23,24,25,27,28,29,31,32,33,34,36,37,38,39,40,41]. There are two that address unusual clinical symptoms and findings [26,35], and there is one that compares how COVID-19 affects patients with cancer and those without cancer [30].

### 3.1. Description of the Publications

The research on COVID-19 is multidisciplinary. The fields of internal medicine (16.37%) and public environmental occupational health (7.82%) are worth mentioning (Figure 3). Medicine is the main area of publication in the field of health, as it is one of the oldest research fields [42]. Similarly, public environmental occupational health has been studied for centuries. However, this research field has significantly increased in the past few years [43].

The quartile according to the SCImago Journal Rank (SJR) is included in the table to show the importance and relevance of the main journals that have published the most articles, a dimension selected based on its quality and the fact that its use is widespread among the international scientific community. Quartiles are based on the rank of each journal within its topic and are measured by assessing the distribution of the impact factor of a given journal for that topic category. SCImago Journal Rank is a website of scientometric and informetric values, which allow researchers to monitor the behavior and impact of their contributions on an international scale; that is, it measures the scientific influence of the journals according to the number of citations. The citations are weighted depending on their field and prestige [44]. Table 2 shows the 10 journals with the largest number of publications.

After comparing our results with the first bibliometric publications on COVID-19, we could observe that the journals with the largest number of publications were as follows: Journal of Medical Virology, which has the most publications (n = 25), followed by Chinese Journal of Tuberculosis and Respiratory Diseases (n = 9), Journal of Travel Medicine (n = 8), Journal of Clinical Medicine (n = 8), Lancet (n = 7), Radiology (n = 6) and JAMA (n = 5) [45].

As shown in Table 3, the authors with the largest number of publications on COVID-19 were Wang Y (0.55%), Mahase E (0.47%) and Li Y (0.43%).

The United States (26.75%), China (14.55%) and Italy (12.32%) were the countries with the highest publication rates (Figure 4).

Similarly, the study by Fan et al. [46] analyzed the countries with the highest rates of publication and found that the authors with a large number of publications on COVID-19 in English-language journals were as follows: Li Y had the highest number of published papers (7, 4.9%), followed by Benvenuto D, Eurosurveillance Editorial Team and Leung GM (5, 3.5%) and Angeletti S, Gao GF, Ran J, Wei Y, Wu JT and Yang G (4, 2.8%). As regards Chinese journals, the authors with the largest numbers of publications were as follows: Wang YG and Yang FW published the highest number of papers (6, 0.8%), followed by Wang YJ (5, 6.9%). Similarly, the study by Belli et al. [47] analyzed the first 12 weeks after the outbreak and found that China represented 32.5% of the publications, followed by the US with 29.44%.

The most frequently used keywords were “COVID-19” (4089 publications), “Coronavirus” (1618 publications) and “SARS-CoV-2” (1488 publications). Figure 5 shows the 30 most used keywords from the most relevant publications. This is consistent with the study conducted by Fan et al. [46], in which the 10 most common keywords in English-language publications were “SARS-CoV-2”, “COVID-19”, “China”, “SARS”, “Epidemic”, “Adult”, “Psychological”, “Nucleic acids”, “Plague” and “Infection”. Moreover, the most common keywords used in Chinese journals were “COVID-19”, “SARS-CoV-2”, “Prevention and control”, “Traditional Chinese Medicine”, “Computed tomography”, “Epidemic”, “Public health”, “MERS”, “Pneumonia” and “Male”.

### 3.2. Clustering Function

By using the clustering function, each publication in the citation network is assigned to a group, which means that publications that are close in the citation network must belong to the same group. Consequently, each of these groups consists of publications that are strongly connected through their citations. In this way, it could be interpreted that every group represents a different topic in the scientific literature. In order to differentiate the groups, each of them has been assigned a specific color. Additionally, the links between groups have been marked using colored lines. The clustering function identified 16 groups, 9 of which have a significant number of articles; however, the remaining groups only account for 1.10% (Figure 6). Table 4 shows the information of the citation networks for the nine main groups, listed from the largest to the smallest according to their size.

In Group 1, 4121 publications and 15544 citations were found throughout the network. The most cited publication is that of Guan et al. [22], published in April 2020 in the New England Journal of Medicine, which is also at the top of the 20 most cited publications. In this group, the different articles analyze the viral transmission of SARS-CoV-2, the most frequent symptoms (fever, cough, diarrhea, etc.) and experimental treatment methods such as chloroquine phosphate (Figure 7).

For Group 2, 2481 publications and 6424 citations were found throughout the network. The most cited publication is that of Liang et al. [30], published in March 2020 in Lancet Oncology. In this paper, the authors discuss how cancer patients are more vulnerable to COVID-19 than people without cancer. This is due to their immunosuppressive status resulting from malignant neoplasm or cancer treatments such as chemotherapy. Furthermore, these patients have a worse prognosis. Therefore, three control strategies were established for cancer patients during the COVID-19 crisis:Postpone surgery and chemotherapy treatment in patients diagnosed with cancer in a stable state.Improve staff protection.Increase surveillance or provide more intensive treatment when cancer patients are infected with SARS-CoV-2, especially in elderly patients or those with comorbidities.

By analyzing the articles belonging to Group 2, it can be noticed that they address how COVID-19 can affect patients with cancer and how they should be treated. Additionally, they discuss the different methods of transmission, namely, through droplets, aerosols or contact with elements such as metal. Consequently, an analysis of protection methods against COVID-19 is offered, both for health professionals and other members of staff, with the final recommendation being to opt for telemedicine.

Additionally, the procedure for performing tracheotomies in the ICU and in the operating room is also explained (Figure 8).

In Group 3, 960 publications and 1990 citations were found throughout the network. The most cited publication is that of Chen et al. [48], published in April 2020 in Lancet Psychiatry. This article analyzed the physical and psychological pressures to which health professionals were exposed. To this end, a psychological response plan was developed, covering three main areas:Development of a medical/psychological intervention team, which provided online courses in order to guide medical staff to deal with common psychological issues.A psychological hotline team, which provided guidance and supervision on solving psychological issues.Psychological interventions, providing group activities aimed at relieving stress.

Moreover, health professionals presented diverse concerns, namely, infecting their relatives, poor knowledge of the disease, lack of protective equipment, inability to treat patients and poor rest. Therefore, hospitals were forced to improve their resting areas, food supply and safety measures. They concluded in this study that preserving the mental health of employees is essential to improving the management of infectious diseases; however, it is still uncertain which approach would be the best. Therefore, the articles belonging to Group 3 address the psychological problems of health professionals, medical students and patients (anxiety, stress or depression). Furthermore, they show methods of treatment aimed at reducing mental health issues, such as support supervisors or online mental health services (questionnaires, books, communication programs, psychological guidance or cognitive therapy) (Figure 9).

In Group 4, 835 publications and 1220 citations were found throughout the network. The most cited publication is that of Shi et al. [35], published in April 2020 in Lancet Infectious diseases. The authors described findings from thorax computed tomography (CT) scans over the course of the disease. Patients were grouped depending on the onset of the symptoms and the results of their first CT scan. As a result, the authors found that COVID-19 pneumonia causes abnormalities that can be found in CT images of the thorax, even in asymptomatic patients, as well as the quick development of ground-glass focal opacities, which evolve from unilateral to bilateral diffuse lesions and advance or coexist with pulmonary consolidations over 1–3 weeks. Therefore, they concluded that combining CT assessment and clinical and laboratory findings could help in providing an early diagnosis of COVID-19 pneumonia.

Articles from this group focus on clinical findings in computerized tomography images from the lungs due to COVID-19 pneumonia (Figure 10).

In Group 5, 524 publications and 1454 citations were found throughout the network. The most cited publication is that of Mao et al. [49], published in April 2020 in JAMA Neurology. In this article, the authors analyze neurological disturbances in patients infected with COVID-19. They examined 214 hospitalized patients, and the neurological disturbances were categorized as follows:Central nervous system manifestations: dizziness, headache, altered states of consciousness, acute cerebrovascular disease, ataxia and seizures.Peripheral nervous system manifestations: taste, sight, smell and nervous alterations.Manifestations of muscular-skeletal lesions.

The study showed that 36.4% of the patients presented neurological alterations. They advised that, upon detecting any neurological disturbance, doctors should consider COVID-19 as a differential diagnosis in order to avoid late or erroneous diagnosis.

When analyzing the total number of articles in this group, it is clear that their main aim is to analyze neurological alterations related to COVID-19 (Figure 11).

In Group 6, 519 publications and 1206 citations were found throughout the network. The most cited publication is that of Emanuel et al. [50], published in May 2020 in the New England Journal of Medicine. This article analyzes the lack of healthcare resources in US hospitals during the COVID-19 crisis, namely, ICU beds, respiratory ventilators, qualified healthcare personnel for critical care units, personal protective equipment and medication. In this sense, they presented recommendations to maximize the benefits (saving the largest number of lives or saving the largest number of years of life by giving priority to patients who will survive longer after treatment), to treat all patients in an equal manner, to encourage and reward instrumental value (the instrumental value could be fostered by giving priority to those who can save others, or rewarding those who have saved others in the past) and to give priority to the most disadvantaged (those who are sick or the youngest people). Articles in this group analyzed how to take advantage of and distribute limited critical care resources during a pandemic. Furthermore, in order to avoid further outbreaks, these authors determined that social distancing could be necessary until 2022 and that continuous surveillance of COVID-19 should be conducted, given that a resurgence of contagion could be possible until 2024 (Figure 12).

In Group 7, 369 publications and 933 citations were found throughout the network. The most cited publication is that of Dong et al. [51], published in March 2020 in JAMA: Journal of the American Medical Association. This article analyzed whether SARS-CoV-2 could be transmitted in the uterus from the infected mother to her baby before birth. To this end, both the mother and baby underwent a thorax CT scan; furthermore, RT-PCR (reverse transcription-polymerase chain reaction) was conducted for SARS-CoV-2 nucleic acid through a nasopharyngeal swab, and IgM (Immunoglobulin M) and IgG (Immunoglobulin G) antibodies, cytokines and other biochemical markers in the blood were analyzed. Additionally, vaginal secretions of the mother during labor were tested with RT-PCR.

The results of the analysis showed that newborns born from a mother with COVID-19 had high antibody levels and unusual cytosine test results two hours after birth.

High IgM antibody levels suggest that the newborn was infected in the uterus since these antibodies are not transferred to the fetus from the mother. Although the possibility of infection at birth cannot be dismissed, IgM antibodies do not usually appear until 3–7 days after the infection. Conversely, IgG antibodies may be transmitted to the fetus through the placenta and may appear at a later time than IgM antibodies, so high IgG levels could be a symptom of maternal or infant infection.

The articles in this group address coronavirus infection in children and babies (Figure 13).

In Group 8, 149 publications and 167 citations were found throughout the network. The most cited publication is that of Luo et al. [52], published in February 2020 in the Chinese Journal of Integrative Medicine. This study consisted of an analysis of Chinese medicine and its potential use for preventing COVID-19. To this end, 23 provinces in China were treated with “Chinese Medicine” to strengthen Qi and protect it from external pathogens by dispersing wind, dissipating heat and reducing humidity. They found that the Chinese herbal formula could be an alternative treatment approach to prevent COVID-19 in a population at high risk. Nevertheless, further studies are necessary to confirm this theory.

In this group, the articles analyze the use of traditional medicine to protect patients from COVID-19. In addition, they also establish that environmental pollution can be a trigger for enhanced transmission of COVID-19 (Figure 14).

In Group 9, 140 publications and 199 citations were found throughout the network. The most cited publication is that of Fang et al. [38], published in April 2020 in Lancet Respiratory Medicine. This article shows the importance of half-face masks for preventing the spread of respiratory infections. Evidence shows that COVID-19 can be transmitted even before the carrier shows any symptoms, and, thus, the transmission rate could be reduced if everyone, including those who are already infected but asymptomatic and yet contagious, wore masks.

These articles emphasize that authorities should optimize the distribution of masks, prioritizing frontline health workers and the needs of the most vulnerable population, as the latter is more susceptible to infection and mortality, including older adults and people with other underlying medical conditions. This group analyzes the different methods for individual protection, such as half masks, safety spectacles, gloves, facial shields, respirators, air purifiers and robes (Figure 15).

After analyzing the relationships among the nine main groups using the drilling down functionality, a connection between Groups 1 and 5 (Figure 16) was found. Furthermore, it was found that publications by the authors An P, Baud D and Chen CJ appear in both groups.

#### 3.2.1. Sub-Clusters in Group 1

Thirteen sub-clusters were found (Figure 17), five of which have a significant number of publications (Table 5). The rest of the groups are relatively small, with fewer than 650 publications and 1072 citation networks.

#### 3.2.2. Sub-Clusters in Group 2

Sixteen sub-clusters were found (Figure 18), only four of which have a significant number of publications (Table 6). The rest of the groups are relatively small, with fewer than 800 publications and 1857 citation networks.

## 4. Discussion

The main databases, such as Web of Science or Scopus, make it possible to create citation networks. However, their usefulness is limited when conducting a systematic review of all of the existing literature on a subject, as they do not offer a general overview of the connections among the citations of a group of publications. For this reason, the CitNetExplorer software was selected, as it allows the researcher to visualize, analyze and explore citation networks of scientific publications. Hence, CitNetExplorer offers a more detailed analysis through the creation of citation networks when compared to other databases, such as Web of Science or Scopus [21].

The main goal of this study was to analyze the existing literature on COVID-19. It should be taken into consideration that the search was carried out before the existence of an effective vaccine against SARS-CoV-2 in humans. In order to identify relevant publications, the Web of Science database was used, which is one of the most comprehensive databases, as its searching range goes back to the year 1900. Nevertheless, it must be taken into consideration that Web of Science only accepts international journals once they have undergone a rigorous selection process. Thus, once the existing bibliography had been downloaded from WOS, the CitNetExplorer software allowed us to collect and analyze every available piece of literature on COVID-19 to date. Furthermore, the connections between fields of study and the different research groups were obtained through the analysis of citation networks. The function called “clustering” was used to collect the results, and the publications were then grouped according to the relationships among the citations. The drilling down function was used to examine the existing bibliography for each group, and the core publication function was used to show the main publications, that is, to show the publications with a minimum number of citations. Therefore, these functions make it possible to conduct a complete study and analysis of the research on the field of study.

The first publication on COVID-19 was published by Bogoch et al. [56] on 14 January 2020 in the Travel Medicine journal. When this article was published, there was news about a pneumonia outbreak in Wuhan, China. Consequently, the authors analyzed the potential for the international dissemination of this disease through air travel. Subsequently, numerous research pieces followed the same idea, including studies on epidemic processes, as well as measures and strategies for prevention and control. In doing so, they analyzed the source of infection, mode of transmission, the population’s susceptibility and other influencing factors. When comparing these articles to previous studies on other viruses, such as SARS-CoV or MERS CoV, they found that the most frequent transmission routes, like respiratory droplets and direct contact, were the same as those for COVID-19 [57,58].

Furthermore, it was found that, even though COVID-19 does not seem as severe as SARS-CoV and MERS-CoV, the sudden increase in the number of cases and growing evidence of human transmission suggest that this virus is more infectious [59,60].

The countries with the largest number of publications are China, the US and Italy. At the beginning of the pandemic, as expected, most of the research works published in international journals were written by Chinese researchers. However, this has been a cause of major concern among frontline health workers and politicians due to the language barrier. Currently, publications mainly cover the symptomatology and focus on finding an effective vaccine against COVID-19. This explains why research is being carried out in this field in countries with higher income and thus better infrastructures, which leads to an increasing number of publications. We suppose that countries with lower income focus on other research fields, such as transmission or epidemiology, although with fewer publications [61,62]. Similarly, the upward trend in numbers of publications in countries such as the US or the UK has been linked to a combination of factors, such as the fact that they are English-speaking countries or the possible connections between different research groups within the scientific community [63,64].

Other bibliometric studies on COVID-19 and SARS-CoV-2, such as those conducted by Lou et al. [65] and Tran et al. [66], have found a significant increase in the number of publications over the last few months. Comparing our bibliographical research to theirs, we found that June was the month in which the number of publications was significantly larger when compared to other months. As more publications become available in the upcoming months, we foresee that these numbers will grow significantly. The reason for this might be that our research scope was wider and comprised more types of documents and a longer period. Furthermore, of the studies in June, the research by McGonagle et al. [67] should be highlighted, as they found that patients with severe pneumonia related to COVID-19 might show signs of systemic hyperinflammation (macrophage activation syndrome) or cytokine storm (secondary hemophagocytic lymphohistiocytosis).

In this sense, the clinical features of COVID-19 are diverse, from the asymptomatic state to acute respiratory distress syndrome and multiple organ dysfunction. The study conducted by Chen et al. [68] showed that the progression of the disease was linked to increased inflammatory cytokines, namely, IL2, IL7, IL10, GCSF, IP10, MCP1, MIP1A and TNFα. Additionally, the study conducted by Casey et al. [69] found that, in some patients, COVID-19 was related to a hypercoagulable state and to an increased risk of venous thromboembolism. In addition, they found that, unlike patients with pulmonary embolism, positive COVID-19 cases rarely showed hemoptysis.

On the other hand, the study conducted by Yang et al. [70] suggested that the gastrointestinal tract and the liver could also be infected by the SARS-CoV-2. This is due to the gastrointestinal epithelial cells and hepatic cells expressing ACE2 enzymes (main receptor of SARS-CoV-2). These findings prove that, even when SARS-CoV-2 has been eliminated from the respiratory tract in some patients, the virus continues replicating through the gastrointestinal tract and could be excreted in the feces. The study by Xu et al. [71] analyzed the characteristics of 10 pediatric patients with COVID-19 and found that 80% of them recurrently tested positive in rectal swabs, even after the nasopharyngeal tests had come back negative.

In turn, the study by Ellul et al. [72] showed a broad range of neurological manifestations in COVID-19 patients. Sixteen out of the 214 patients in Chinese hospitals and 40 out of the 58 patients in French hospitals showed signs of encephalopathy. Moreover, Guillain-Barré syndrome was found in 19 of the patients. On the other hand, anosmia and ageusia were frequent and could occur in the absence of other clinical features, while 2–6% of hospitalized patients showed apoplexy.

Regarding the treatment, various antiviral medications have been tested, such as ribavirin and lopinavir/ritonavir, which are commonly used in the treatment and prevention of AIDS. Thus, the study showed that a patient treated with lopinavir/ritonavir together with ribavirin had better results when compared to those to whom only ribavirin was administered [68]. Chloroquine has also been used as a form of treatment and has shown antiviral effects on the cells of primates infected by SARS-CoV. Similarly, in the research by Vincent et al. [73], a favorable inhibition of the virus spread was observed when the cells were treated with chloroquine before or after SARS-CoV infection.

It should be noted that the use of remdesivir was authorized on 3 July as the first conditionally marketable treatment. This facilitates early access to medication in emergency public health situations. This antiviral is a treatment against COVID-19 for adults and teenagers aged 12 or over who suffer from pneumonia and who require supplementary oxygen [74]. On the other hand, there are currently 110 possible vaccines from different laboratories, of which 102 are in a preclinical phase and 8 are in the clinical phase [3]. Research conducted by Gao et al. [75] was published recently, and it reports a pilot-scale production of PiCoVacc, a possible vaccine that would inactivate the SARS-CoV-2 virus. This vaccine has created antibodies in mice, rats and non-human primates. They also neutralized 10 strains representative of SARS-CoV-2, which might suggest a greater possible neutralizing capacity against other strains. This leads us to believe that there is a strong probability of developing an effective vaccine for humans shortly. Additionally, upon development/availability and approval of a vaccine or a new effective treatment against SARS-CoV-2, we believe it necessary to carry out a new search and citation analysis.

The future of COVID-19 is still uncertain. This is due to the lack of knowledge of the clinical progression of this illness. In particular, many patients experience a recurrence of symptoms after periods of showing no signs, or they show symptoms such as febricula, asthenia or headaches for a longer period. Moreover, some patients test positive again in the PCR test after testing negative. This might be due to the sensitivity of this test, especially when the viral charge is lower and/or when the viral RNA is on the threshold of the detection rate of the test [76]. On the other hand, whether COVID-19 can also have after-effects is still to be determined. The research by Zhang et al. [77] found that 4.6% of survivors showed signs of pulmonary fibrosis. We recognize that there is still further research to be conducted in order to confirm whether COVID-19 can have after-effects or whether it varies depending on the severity of the condition.

Consequently, the number of studies on COVID-19 is constantly on the increase since more research is needed in order to fight this illness. Citation network studies are becoming more numerous, as this is the only analysis method that provides a global overview of the different research fields within a specific topic. Furthermore, CitNetExplorer software facilitates the analysis of all existing research on a specific topic through detailed studies. This could change how studies in different research areas are conducted.

## 5. Conclusions

In conclusion, this research offers a specific and objective analysis of the main articles on COVID-19 and SARS-CoV-2. In addition, it was possible to visualize, analyze and explore the most cited articles and citation networks existing to date using the Web of Science database and the Citation Network Explorer database.

By doing so, we have obtained information regarding the epidemiology, clinical features, diagnosis and treatment of COVID-19. The importance of neurological and gastrointestinal alterations in a high percentage of positive cases of COVID-19 is also worth mentioning. In addition to the great progress made towards effective treatment, the research conducted by Gao et al. must be highlighted, as it shows the first results regarding the creation of the first vaccine for humans.

Therefore, COVID-19 disease is a relevant field for researchers, with the number of publications continuously on the increase. Consequently, the clinical progression of the disease will be understood in the upcoming months, and a new vaccine and an effective treatment will be discovered soon.

In this way, this paper contributes to a better understanding of the information structure by identifying, in chronological order, the pieces of knowledge on different aspects of COVID-19 that are interconnected. Furthermore, this paper explores researchers’ capacity to generate scientific knowledge on international health crises in order to advance our understanding of this pandemic and control its effects.

## Figures and Tables

**Figure 1 ijerph-17-07690-f001:**
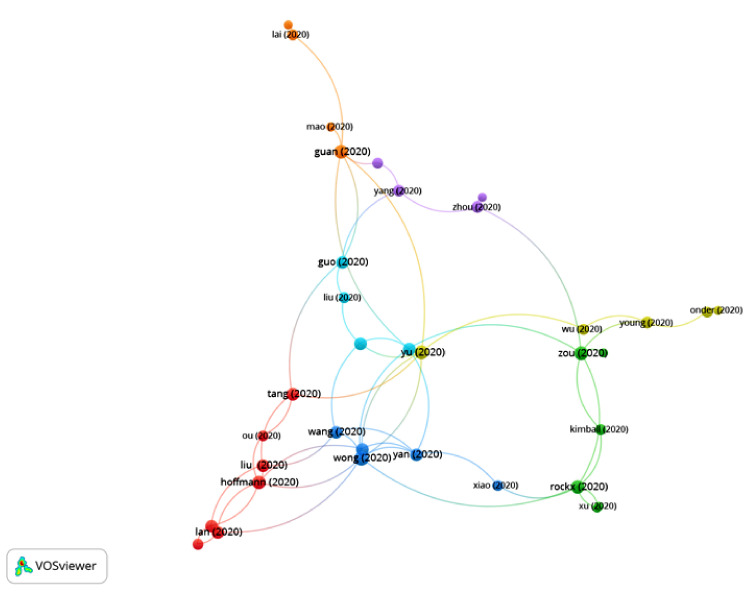
Citation networks on COVID-19.

**Figure 2 ijerph-17-07690-f002:**
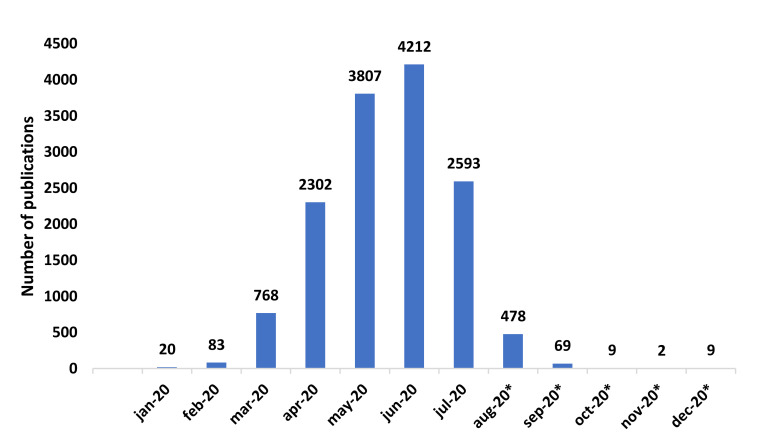
Number of publications per month. * Expected date of publication.

**Figure 3 ijerph-17-07690-f003:**
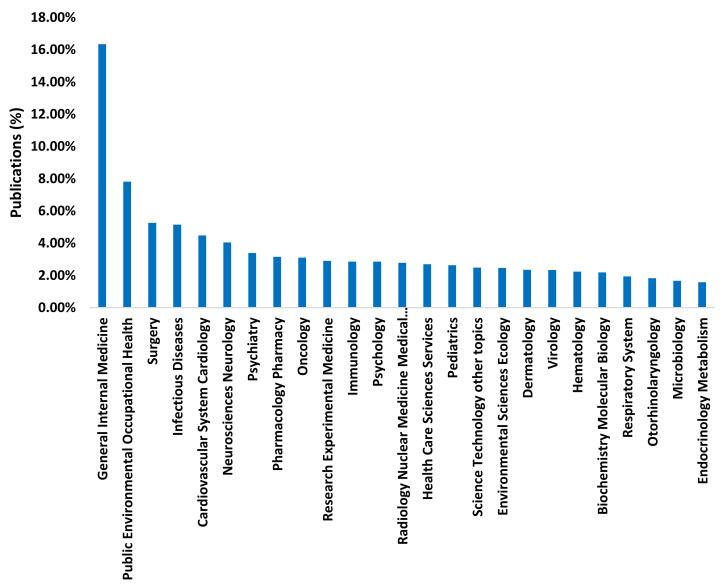
Percentage of publications by research area.

**Figure 4 ijerph-17-07690-f004:**
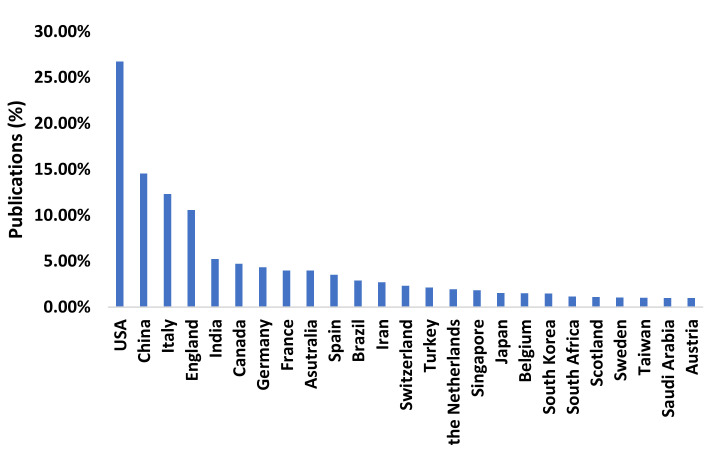
Publication rate by country.

**Figure 5 ijerph-17-07690-f005:**
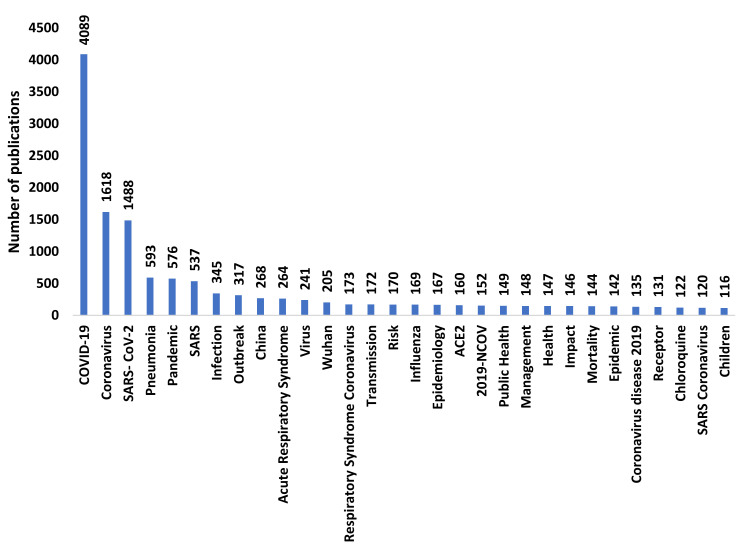
The most common keywords.

**Figure 6 ijerph-17-07690-f006:**
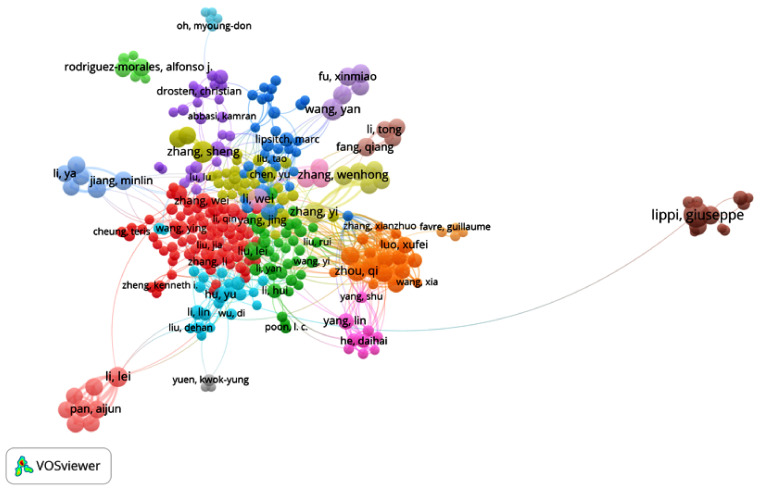
Clustering function in the citation network on COVID-19.

**Figure 7 ijerph-17-07690-f007:**
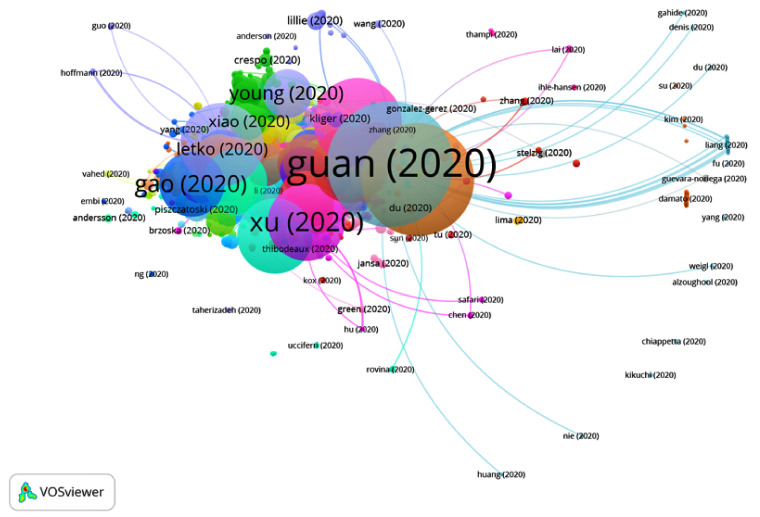
Citation network for Group 1.

**Figure 8 ijerph-17-07690-f008:**
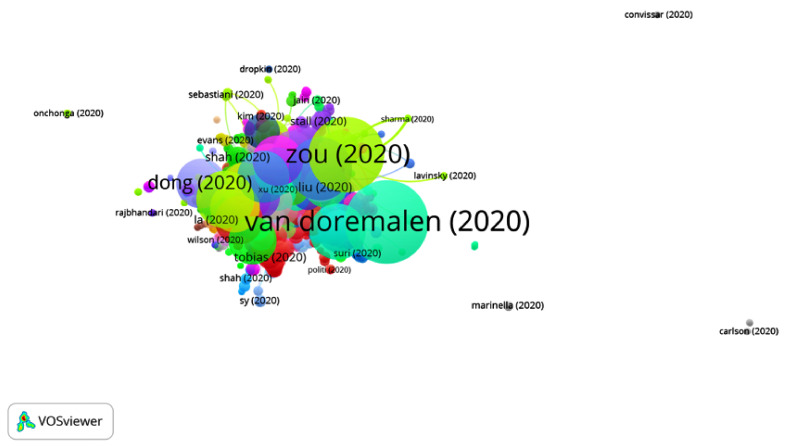
Citation network for Group 2.

**Figure 9 ijerph-17-07690-f009:**
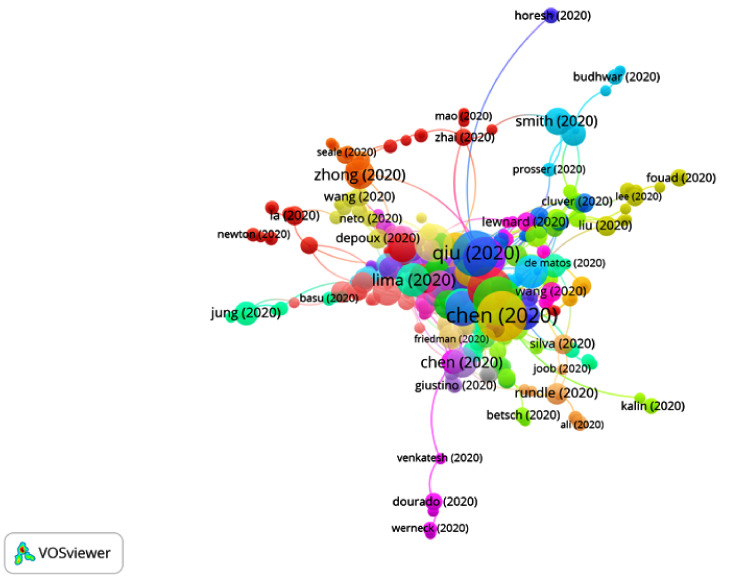
Citation network for Group 3.

**Figure 10 ijerph-17-07690-f010:**
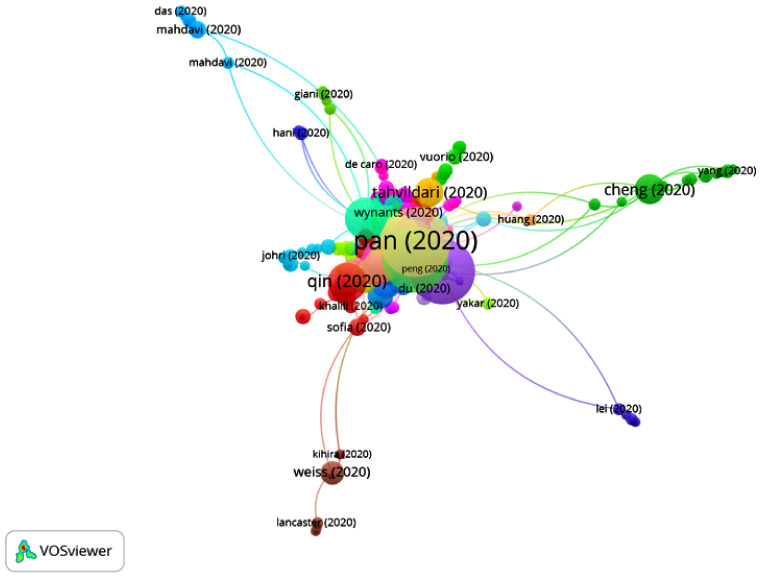
Citation network for Group 4.

**Figure 11 ijerph-17-07690-f011:**
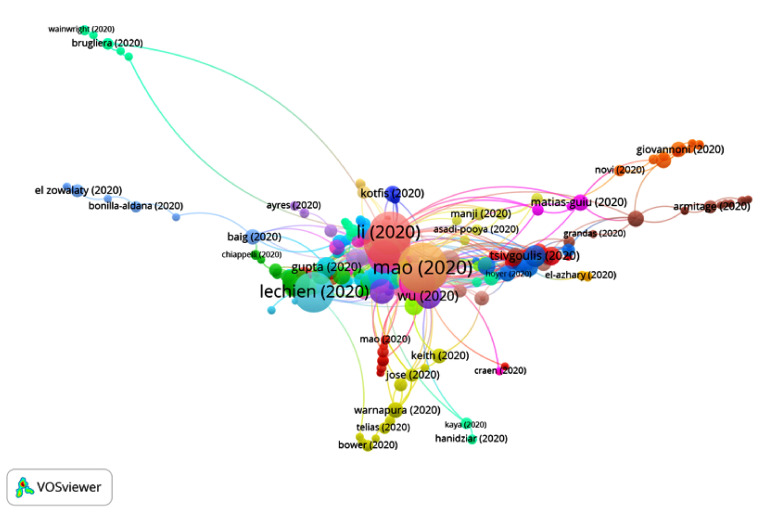
Citation network for Group 5.

**Figure 12 ijerph-17-07690-f012:**
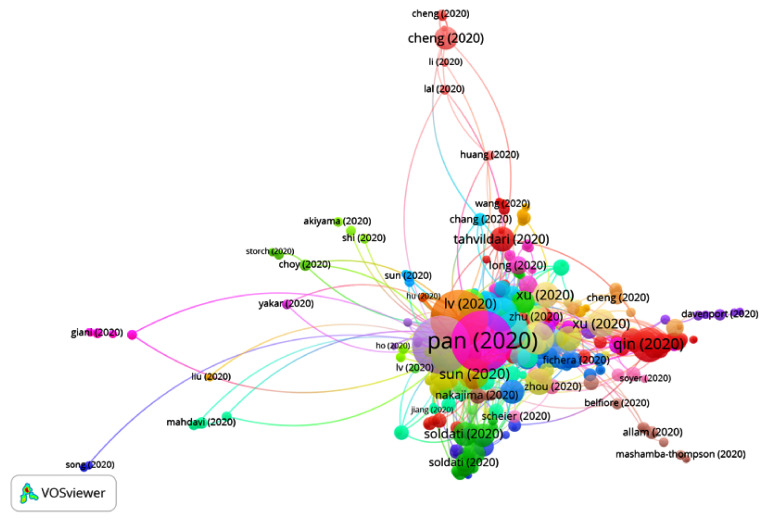
Citation network for Group 6.

**Figure 13 ijerph-17-07690-f013:**
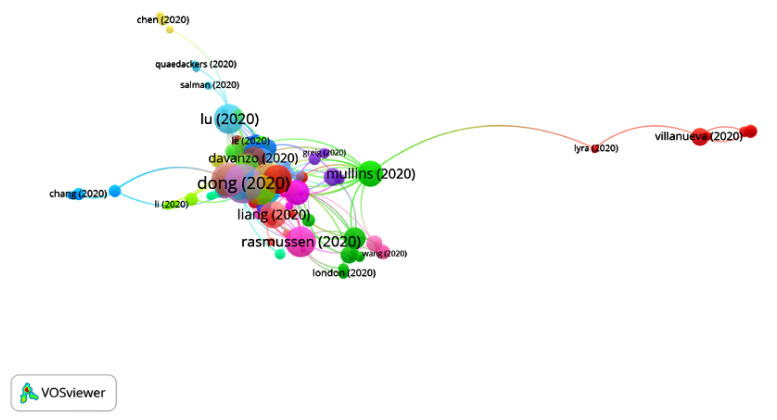
Citation network for Group 7.

**Figure 14 ijerph-17-07690-f014:**
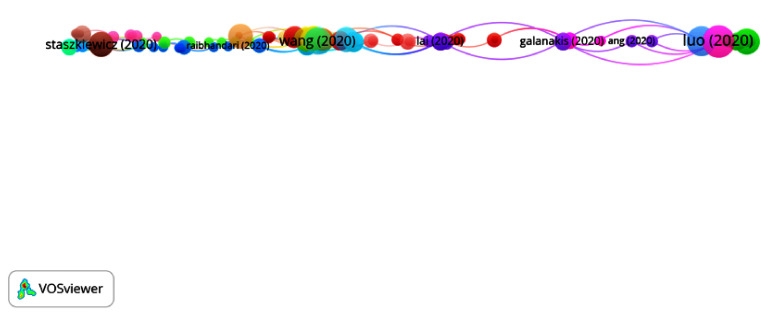
Citation network for Group 8.

**Figure 15 ijerph-17-07690-f015:**
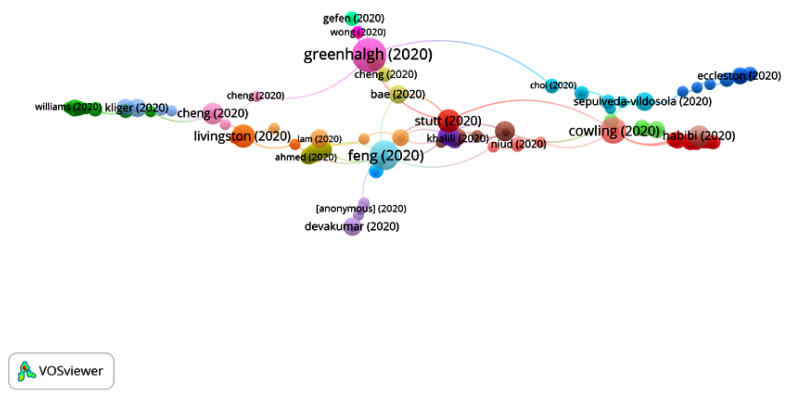
Citation network for Group 9.

**Figure 16 ijerph-17-07690-f016:**
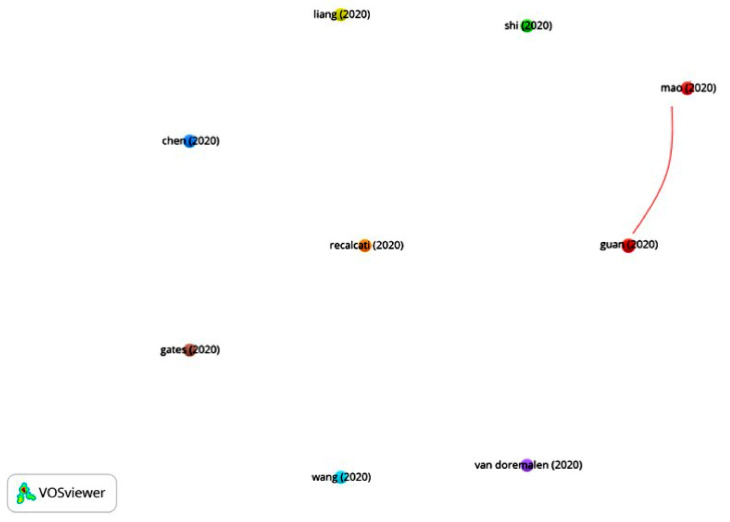
Similarities between the nine main groups.

**Figure 17 ijerph-17-07690-f017:**
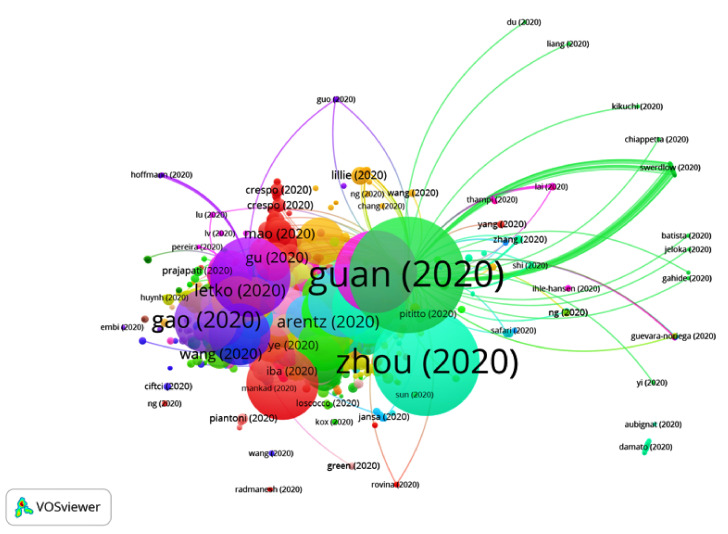
Citation network from the sub-clusters in Group 1.

**Figure 18 ijerph-17-07690-f018:**
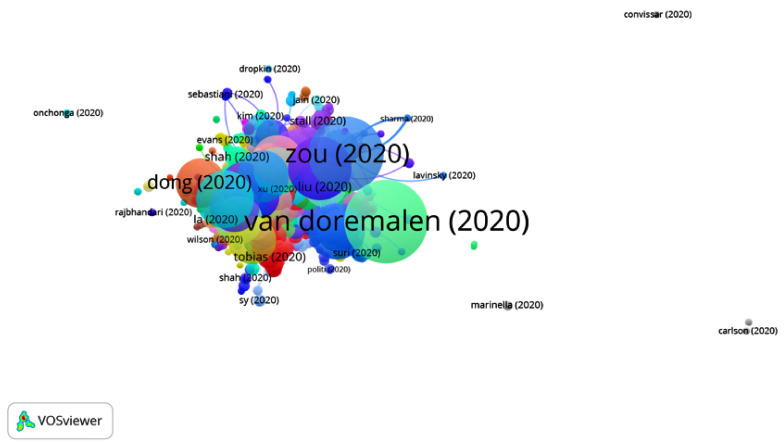
Citation network from the sub-clusters of Group 2.

**Table 1 ijerph-17-07690-t001:** Description of the 20 most cited publications on COVID-19.

Author	Title	Journal	Year	Citation Index	H-Index
Guan et al. [22]	Clinical Characteristics of Coronavirus Disease 2019 in China	New England Journal of Medicine. 2020 Apr; 382(18):1708–1720	2020	1150	1
Zhou et al. [23]	Clinical course and risk factors for mortality of adult inpatients with COVID-19 in Wuhan, China: a retrospective cohort study	Lancet. 2020 Mar; 395(10229):1054–1062	2020	793	1
Wu et al. [24]	Characteristics of and Important Lessons From the Coronavirus Disease 2019 (COVID-19) Outbreak in China: Summary of a Report of 72 314 Cases From the Chinese Center for Disease Control and Prevention	JAMA—Journal of the American Medical Association. 2020 Feb; 322(13):1239–1242	2020	706	1
Hoffmann et al. [25]	SARS-CoV-2 Cell Entry Depends on ACE2 and TMPRSS2 and Is Blocked by a Clinically Proven Protease Inhibitor	Cell. 2020 Apr; 181(2):271–280.e8	2020	366	1
Xu et al. [26]	Pathological findings of COVID-19 associated with acute respiratory distress syndrome	Lancet Respiratory Medicine. 2020 Apr;8(4):420–422	2020	346	2
Yang et al. [27]	Clinical course and outcomes of critically ill patients with SARS-CoV-2 pneumonia in Wuhan, China: a single-centered, retrospective, observational study	Lancet Respiratory Medicine. 2020 May;8(5):475–481	2020	346	1
van Doremalen et al. [28]	Aerosol and Surface Stability of SARS-CoV-2 as Compared with SARS-CoV-1	New England Journal of Medicine. 2020 Apr;382(16):1564–1567	2020	339	2
Mehta et al. [29]	COVID-19: consider cytokine storm syndromes and immunosuppression	Lancet. 2020 Mar; 395(10229):1033–1034	2020	331	1
Liang et al. [30]	Cancer patients in SARS-CoV-2 infection: a nationwide analysis in China	Lancet Oncology. 2020 Mar;21(3):335–337	2020	269	1
Gao et al. [31]	Breakthrough: Chloroquine phosphate has shown apparent efficacy in treatment of COVID-19 associated pneumonia in clinical studies	Bioscience Trends. 2020 Mar; 14(1):72–73	2020	237	1
Cao et al. [32]	A Trial of Lopinavir–Ritonavir in Adults Hospitalized with Severe Covid-19	New England Journal of Medicine. 2020 May; 382(19):1787–1799	2020	221	1
Lai et al. [33]	Severe acute respiratory syndrome coronavirus 2 (SARS-CoV-2) and coronavirus disease-2019 (COVID-19): The epidemic and the challenges	International Journal of Antimicrobial Agents. 2020 Mar;55(3):105924	2020	203	1
Bai et al. [34]	Presumed Asymptomatic Carrier Transmission of COVID-19	JAMA—Journal of the American Medical Association. 2020 Feb 21;323(14):1406–1407	2020	189	1
Shi et al. [35]	Radiological findings from 81 patients with COVID-19 pneumonia in Wuhan, China: a descriptive study	Lancet Infectious Diseases. 2020 Apr;20(4):425–434	2020	186	1
Wang et al. [36]	Detection of SARS-CoV-2 in Different Types of Clinical Specimens	JAMA—Journal of the American Medical Association. 2020 Mar 11;323(18):1843–1844	2020	176	1
Gorbalenya et al. [37]	The species Severe acute respiratory syndrome-related coronavirus: classifying 2019-nCoV and naming it SARS-CoV-2	Nature Microbiology. 2020 Apr;5(4):536–544	2020	172	1
Fang et al. [38]	Are patients with hypertension and diabetes mellitus at increased risk for COVID-19 infection?	Lancet Respiratory Medicine. 2020 Apr;8(4):e21	2020	153	1
Pan et al. [39]	Time Course of Lung Changes at Chest CT during Recovery from Coronavirus Disease 2019 (COVID-19)	Radiology. 2020 Jun;295(3):715–721	2020	152	1
Remuzzi et al. [40]	COVID-19 and Italy: what next?	Lancet. 2020 Apr;395(10231):1225–1228	2020	149	1
Onder et al. [41]	Case-Fatality Rate and Characteristics of Patients Dying in Relation to COVID-19 in Italy	JAMA—Journal of the American Medical Association. 2020 Mar: 323(18):1775–1776	2020	144	1

**Table 2 ijerph-17-07690-t002:** Top 10 journals with the most publications.

Journal	Total Publications COVID-19	Impact Factor	Quartile Score	SJR (Scimago Journal & Country Rank) (2019)	Cites/ Docs (2 Years)	Total Cites (2019)	H Index	Country
BMJ: British Medical Journal	633	30.22	Q1	2.05	4.23	16,584	412	United Kingdom
Cureus	204	-	-	-	-	-	-	United States
International Journal of Environmental Research and Public Health	166	2.85	Q1	0.74	3.18	18,252	92	Switzerland
Lancet	166	60.39	Q1	14.55	44.86	60,350	747	United Kingdom
Psychological Trauma: Theory Research Practice and Policy	145	2.59	Q2	1.41	3.19	1052	42	United States
Critical Care	144	6.41	Q1	2.38	5.59	5514	160	United Kingdom
Journal of Infection	138	4.84	Q1	1.98	4.97	1977	96	United Kingdom
New England Journal of Medicine	114	74.70	Q1	18.29	40.15	74,865	987	United States
International Journal of Infectious Diseases	107	3.20	Q1	1.44	3.33	2692	79	The Netherlands
JAMA Journal of the American Medical Association	102	45.54	Q1	5.91	11.38	34,346	654	United States

**Table 3 ijerph-17-07690-t003:** Top 10 authors with the largest number of publications.

Author	Number of Publications	H Index	Total Citations	Citation Average
Wang Y	84	16	1399	16.65
Mahase E	72	4	88	1.22
Li Y	65	8	509	7.83
Li L	62	9	2418	39.00
Wang J	57	13	1077	18.89
Iacobucci G	56	3	32	0.57
Liu J	56	12	2529	45.16
Zhang L	52	10	849	16.33
Zhang Y	52	14	2418	46.50
Liu Y	49	17	4710	96.12

**Table 4 ijerph-17-07690-t004:** Information on the citation network of the 9 main groups.

Main Cluster	Number of Publications	Number of Citation Links	Number of Citations Median (Range)	Number of Publications with ≥4 Citations	Number of Publications in the 100 Most Cited Publications
Group 1	4121	15,544	0 (706–0)	2214	82
Group 2	2481	6424	0 (269–0)	439	5
Group 3	960	1990	0 (57–0)	459	1
Group 4	835	1220	0 (186–0)	306	4
Group 5	524	1454	0 (100–0)	192	5
Group 6	519	1206	0 (89–0)	275	2
Group 7	369	933	1 (55–0)	137	1
Group 8	149	167	0 (16–0)	141	0
Group 9	140	199	0 (22–0)	139	0

**Table 5 ijerph-17-07690-t005:** Main citation network groups from the sub-cluster of Group 1.

Sub-Cluster	1	2	3	4	5
No. of publications	1429	1143	402	234	231
No. of citation links	4875	3810	863	555	553
Most cited publication	Guan et al. [22]	Hoffmann et al. [25]	Mehta et al. [29]	Xiao et al. [53]	Tang et al. [54]
Main Keywords	Covid-19, outbreak, pandemic	Sars-cov-2, virus, Sars	Cytokine storm, inflammation, treatment	Sars coronavirus, pathogenesis, receptor	Coronavirus, transmission, surgery
Topic of discussion	Symptoms and signs in COVID-19 patients.	Analyze how the Sars-Cov-2 can pass through human barriers and can infect different types of cells from different species.	Treatment methods	Alterations in the digestive system as a result of COVID-19	Blood clotting alterations in COVID-19 patients.
Conclusion	Clinical findings can help not only to identify the cause of death but also to provide new ideas on the pneumonia’s pathogenesis related to SARS-CoV-2. This may help doctors to give an appropriate response or to provide a therapeutic strategy for critical patients and thus reduce the mortality rate.	SARS-CoV-2 uses SARS-CoV’s ACE2 receptor for the input and the TMPRSS2 serine protease for the priming of protein S.	Treatment using cytokines and/or tocilizumab is likely to become one of the effective approaches for critically ill COVID-19 patients.	It has been shown that digestive symptoms are common in COVID-19 patients. Furthermore, these patients showed longer blood clotting time and increased hepatic enzyme levels. However, more sample studies are needed to confirm these findings.	COVID-19 patients show signs of venous thromboembolism. It has been proven that an anticoagulant therapy, mainly low molecular weight heparin, seems effective. However, more studies are necessary to establish the type of medication, the dose and the optimal duration of the prophylaxis.

**Table 6 ijerph-17-07690-t006:** Main citation network groups from the sub-cluster of Group 2.

Sub-Cluster	1	2	3	4
No. of publications	509	399	223	176
No. of citation links	1028	830	302	270
Most cited publication	van Doremalen et al. [28]	Lauer et al. [55]	Remuzzi et al. [40]	Lai et al. [33]
Main Keywords	Acute respiratory syndrome, infections, chloroquine	Public health, epidemic, quarantine	Management, personal protective equipment, safety	Epidemiology, sars-cov-2, syndrome
Topic of discussion	Evaluate the presence of SARS-CoV-2 on different surfaces.	Evaluate the incubation period of COVID-19 and its effects on public health.	Evaluate the efficiency of the contention methods for reducing COVID-19 outbreaks.	COVID-19 epidemiology
Conclusion	The presence of SARS-CoV-2 on different surfaces is very similar to that of SARS-CoV-1. The epidemiological differences between these viruses may be due to the high viral loads in the respiratory tract and the possibility that SARS-CoV-2-infected people could transmit the virus while being asymptomatic.	The mean period of incubation is of 5 days, and in most of the people, the symptoms appear within 11 days. However, these numbers vary between mild cases and severe cases.	Containment methods, such as social distancing or the use of masks, have proven to be effective in limiting virus transmission. Therefore, the general public must be aware of the recommendations regarding COVID-19 contention.	Public health authorities must keep tracking the situation to obtain new information about the virus and its outbreaks in order to estimate the risk of further outbreaks with greater precision.

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
