# Peer review of "Citation Network Analysis of the Novel Coronavirus Disease 2019 (COVID-19)"

_ijerph, 2020, doi:10.3390/ijerph17207690_

Round 1
Reviewer 1 Report
The manuscript describes the scientometric evaluation of the recent literature about SARS-CoV-2 and COVID-19. A number of drawbacks are noted hamering the quality and soundness of the findings. The approach of the authors is too large. A number of fields of interest in the cloud of knowledge related to COVID-19 are compared, for example, but in practical terms, considerinds the requirements of different journals for publication in aspects like masks for biosafety of clinical trials, it is impossible to compare these fairly. Most of these findings are thus meaningless.
Maybe it is not also to do a study like these, this type of studies are much more realiable and useful when performed some time after the epicentre of the events, and we are probably and first wave of bothe the pandemics and the body knowledge generated from this. In the present form and present time the manuscript does contribute to the overall situation.
Author Response
Response to Reviewer 1 Comments
Point 1: The manuscript describes the scientometric evaluation of the recent literature about SARS-CoV-2 and COVID-19. A number of drawbacks are noted hamering the quality and soundness of the findings. The approach of the authors is too large. A number of fields of interest in the cloud of knowledge related to COVID-19 are compared, for example, but in practical terms, considerinds the requirements of different journals for publication in aspects like masks for biosafety of clinical trials, it is impossible to compare these fairly. Most of these findings are thus meaningless. Maybe it is not also to do a study like these, this type of studies are much more realiable and useful when performed some time after the epicentre of the events, and we are probably and first wave of bothe the pandemics and the body knowledge generated from this. In the present form and present time the manuscript does contribute to the overall situation.
Response 1: We really appreciate your comments. The bibliometric studies on the coronavirus which have been published so far obtained data mainly from WoS, Scopus and Medline. These are based on a few articles and conduct a traditional bibliometric analysis (authors, collaborations, countries, organisations, journals, etc.). There are only a few bibliometric studies on coronavirus. The first one was conducted by Chiu et al. (2004), and comprised 256 indexed articles in Science Citation Index (SCI) for the period March-July 2003. This study analysed traditional indicators (author, collaboration, journals, language, type of document, organisation, etc.). Since then, other similar studies have been published, such as an analysis of 3202 articles about SARS in Medline (Yang and Yang 2005), 2874 articles about SARS indexed in SCI for the period 1998-2008 (Kostoff and Morse 2011), 443 articles about MERS published in PubMed between 2012 and 2015 (Wang et al. 2016), 883 articles about MERS in Scopus between 2012 and 2015 (Zyoud 2016) and 8619 articles in Scopus about several infectious pathologies (Sweileh 2017). A short letter to the editor has been published this year (Bonilla-Aldana et al. 2020) which provides a more detailed description of the 18,158 articles about coronavirus published in SCI, Scopus and PubMed. Furthermore, with a broader perspective, Zhang et al. (2020) analysed the reaction of the field of scientific research to six international public health emergencies since 2000 (SARS, influenza A (H1N1), Ebola, Zika virus, COVID-19) and they also presented a preliminary analysis of scientific research on COVID-19 based on 3069 indexed papers published in WoS, PubMed and CNKI before 9th April 2020. Moreover, ten bibliometric studies have been published over the past few months which have focused on different aspects of COVID-19.
The main objective of this study, which comprises a larger number of articles, is to analyse the citations network on COVID-19 to date. This type of studies aim at offering a visual representation of the links between articles and authors (who is citing who) to help the researcher to understand the structure of the information by identifying associated pieces of knowledge in chronological order. In this way, this study provides an exhaustive and objective analysis of the main works on COVID-19 and SARS-CoV-2. Furthermore, by using the Web of Science database and the Software Citation Network Explorer, it was possible to visualise, analyse and explore the most cited articles and existing citation networks to date.
The data obtained in this study will be useful to assess the publications on COVID-19 from all over the world and determine the main characteristics of these scientific publications. This paper explores researchers’ capacity to generate scientific knowledge on international health crises, as well as their general ability to collaborate in this task. In order to advance in our knowledge of this pandemic and control its effects, it is essential for researchers to collaborate at the international level and have open and immediate access to scientific publications. Moreover, this paper identifies the countries with the largest number of scientific publications, the extent of their scientific collaboration and their impact.
Consequently, the conclusions have been amended and now explain how this publication could contribute to the dissemination of scientific knowledge and help researchers in future projects:
- 515-519: In this way, this paper contributes to a better understanding of the information structure by identifying, in chronological order, the pieces of knowledge on the different aspects of COVID-19 which are interconnected. Furthermore, this paper explores researchers’ capacity to generate scientific knowledge on international health crises in order to advance in our understanding of this pandemic and control its effects.
Reviewer 2 Report
- abstract line 12, it is not a good idea to mention “first occurred in China” because to date there are a couple of evidences which suspect COVID-19 occurred sporadically in the world in 2019.
- introduction line 33, “was notified in Wuhan, in the Chinese province of Hubei, on the 8th of December 2019”, give reference or origin of this statement.
- figure 1 legend, should indicate the meaning of different colors, and why only just these papers are showed.
- line 138, the junction of two sentences “… with a citation index of 1,150. 1,099 patients …” makes me confused. Could you make them easy to read?
- in the beginning of line 178, please introduce Clustering Function briefly, like what characters to distinguish the different groups, the relationship between the publications and citations in one group, and so on.
- table 4, first column, regulate the font size to fit the cells.
- line 213, delete the –.
- Line 288, “newborns born to a mother with COVID-19”, do you mean newborns born from a mother with COVID-19? Please double check and use a precise expression.
Author Response
Response to Reviewer 2 Comments
Point 1: abstract line 12, it is not a good idea to mention “first occurred in China” because to date there are a couple of evidences which suspect COVID-19 occurred sporadically in the world in 2019.
Response 1: Thank you very much for your comments on how to improve the manuscript. Line 12 has been modified in the manuscript as you recommend:
- 11-12: Background: The first outbreaks of the new coronavirus disease, named COVID-19, occurred at the end of December 2019.
Point 2: introduction line 33, “was notified in Wuhan, in the Chinese province of Hubei, on the 8th of December 2019”, give reference or origin of this statement.
Response 2: Thanks for the recommendation. We have included the reference:
- 33: In this regard, a group of pneumonia cases with an unknown origin was notified in Wuhan, in the Chinese province of Hubei, on the 8th of December 2019 [4].
Point 3: figure 1 legend, should indicate the meaning of different colors, and why only just these papers are showed.
Response 3: Thank you for your comment. In the diagram showing the network, the elements are represented by their label and, by default, by a circle as well. The size of the label and the circle for a given article will depend on its weight. Articles with more weight are represented by larger labels and circles. The label might not be shown for some elements so as to prevent these labels from overlapping. The colour of the article is determined by the group to which the article belongs. The lines between the elements represent links. Figure 1 has been described as follows:
- 129-131: Figure 1 shows the publications with a greater weight and the group to which they belong. The colour of an article represents its group and the lines that connect the elements represent links.
Point 4: line 138, the junction of two sentences “… with a citation index of 1,150. 1,099 patients …” makes me confused. Could you make them easy to read?
Response 4: Thanks again for your suggestion. We have changed the sentence following your advice:
- 140-141: ……. with a citation index of 1,150. In this publication, 1,099 patients were included from 19th December……
Point 5: in the beginning of line 178, please introduce Clustering Function briefly, like what characters to distinguish the different groups, the relationship between the publications and citations in one group, and so on.
Response 5: Following your suggestion, we have briefly explained Clustering Function:
- 214-219: By using the clustering function, each publication in the citation network is assigned to a group, which means that the publications which are close in the citation network must belong to the same group. Consequently, each of these groups consists of publications which are strongly connected as regards their citations. In this way, it could be interpreted that every group represents a different topic in the scientific literature. In order to differentiate the groups, each of them has been assigned a specific colour. Likewise, the links between groups have been marked using coloured lines.
Point 6: table 4, first column, regulate the font size to fit the cells.
Response 6: Thanks for the appreciation. We have modified the size of the cells:
|
Main Cluster |
Number of Publications |
Number of Citation Links |
Number of Citations Median (Range) |
Number of Publications with ≥4 Citations |
Number of Publications in the 100 Most Cited |
|
Group 1 |
4121 |
15544 |
0 (706-0) |
2214 |
82 |
|
Group 2 |
2481 |
6424 |
0 (269-0) |
439 |
5 |
|
Group 3 |
960 |
1990 |
0 (57-0) |
459 |
1 |
|
Group 4 |
835 |
1220 |
0 (186-0) |
306 |
4 |
|
Group 5 |
524 |
1454 |
0 (100-0) |
192 |
5 |
|
Group 6 |
519 |
1206 |
0 (89-0) |
275 |
2 |
|
Group 7 |
369 |
933 |
1 (55-0) |
137 |
1 |
|
Group 8 |
149 |
167 |
0 (16-0) |
141 |
0 |
|
Group 9 |
140 |
199 |
0 (22-0) |
139 |
0 |
Point 7: line 213, delete the –.
Response 7: Thanks for the appreciation. We have deleted the “–” in the manuscript.
Point 8: Line 288, “newborns born to a mother with COVID-19”, do you mean newborns born from a mother with COVID-19? Please double check and use a precise expression.
Response 8: Thanks for for your suggestion. We have changed the expression:
L.332:.. that newborns born from a mother with COVID-19…..
Reviewer 3 Report
The author needs to improve the result part, focusing on the graphs and tables results comparing with previous studies.
Also, I recommend introducing in the conclusion of how these results are useful for the scientific community.
I suggest to revise and to add this study:
Belli, S., Mugnaini, R., Baltà, J., & Abadal, E. (2020). Coronavirus mapping in scientific publications: when science advances rapidly and collectively, is access to this knowledge open to society?. Scientometrics
Author Response
Response to Reviewer 3 Comments
Point 1: The author needs to improve the result part, focusing on the graphs and tables results comparing with previous studies.
Response 1: Thank you very much for your comments on how to improve the manuscript. We have modified the results section:
- 157- 212: 3.1. Description of the publications
The research area on COVID-19 is multidisciplinary. The field of internal medicine (16.37%) and Public Environmental Occupational Health (7.82%) are worth mentioning (figure 3). Medicine is the main area of publication in the field of health as it is one of the oldest research fields [42]. Likewise, Public Environmental Occupational Health has been studied for centuries. However, this research field has significantly increased in the past few years [43].
The quartile according to the Scimago Journal Rank (SJR) has been included in the table to show the importance and relevance of the main journals which have published the most articles, a dimension selected based on its quality and the fact that its use is widespread among the international scientific community. Quartiles are based on the rank of each journal within its topic and are measured by assessing the distribution of the impact factor of a given journal for that topic category. The Scminago Journal Rank is a website of scientometric and informetric values which allow researchers to monitor the behaviour and impact of their contributions on an international scale, that is, it measures the scientific influence of the journals according to the number of citations. The citations are weighed depending on their field and prestige [44]. Table 2 shows the 10 journals with the largest number of publications.
After comparing our results with the first bibliometric publications on COVID-19, we could observe that the journals with the largest number of publications were as follows: Journal of Medical Virology, which has the most publications (n=25), followed by Chinese Journal of Tuberculosis and Respiratory Diseases (n=9), Journal of Travel Medicine (n=8), Journal of Clinical Medicine (n=8), Lancet (n=7), Radiology (n=6), and JAMA (n=5) [45].

Reviewer 4 Report
Comments:
In this study authors aims to identify the different research areas and to determine the most frequently cited publication. Similarly, they aims to analyze the relationships between the publications and the different research groups using the CitNetExplorer software, which examines the development of the scientific literature in a given research field. Finally this research offers a specific and objective analysis of the main articles on COVID- 19 and SARS-CoV-2. On the other hand, it was possible to visualize, analyze and explore the most cited articles and citation networks existing to date using the Web of Science database and the Citation Network Explorer database. The manuscript is very well written, and it has a nice flow of information. The scientific merits of this manuscript are high.
The figures 1, 6, 7, 8, 9, 10, 11, 12, 13, 14, 15, 16, 17, 18 are looks fuzzy; authors could try to increase the resolution of the images.
Author Response
Response to Reviewer 4 Comments
Point 1: In this study authors aims to identify the different research areas and to determine the most frequently cited publication. Similarly, they aims to analyze the relationships between the publications and the different research groups using the CitNetExplorer software, which examines the
development of the scientific literature in a given research field. Finally this research offers a specific and objective analysis of the main articles on COVID- 19 and SARSCoV- 2. On the other hand, it was possible to visualize, analyze and explore the most cited articles and citation networks existing to date using the Web of Science database and the Citation Network Explorer database.
The manuscript is very well written, and it has a nice flow of information. The scientific merits of this manuscript are high.
The figures 1, 6, 7, 8, 9, 10, 11, 12, 13, 14, 15, 16, 17, 18 are looks fuzzy; authors could try to increase the resolution of the images.
Response 1: Thanks for your comment. We have increased the resolutions of the images.